

# Analysis of the effects of biases in ESP forecasts on electricity production in hydropower reservoir management

Richard Arsenault[1,2], Pascal Côté[2]

[1]Department of construction engineering, École de technologie supérieure, Montreal, H3C 1K3, Canada
5  [2]Quebec Power Operations, Rio Tinto, Jonquière, G7S 4R5, Canada

*Correspondence to*: Richard Arsenault (richard.arsenault@etsmtl.ca)

**Abstract.** This paper presents an analysis of the effects of biased Extended Streamflow Prediction (ESP) forecasts on three deterministic optimization techniques implemented in a simulated operational context with a rolling horizon testbed for managing a cascade of hydroelectric reservoirs and generating stations in Québec, Canada. The observed weather data was 10  fed to the hydrological model and the synthetic streamflow thus generated was considered as a proxy for the observed inflow. A traditional, climatology-based ESP forecast approach was used to generate ensemble streamflow scenarios, which were used by three reservoir management optimization approaches. Both positive and negative biases were then forced into the ensembles by multiplying the streamflow values by constant factors. The optimization method's response to those biases was measured through the evaluation of the average annual energy generation in a forward-rolling simulation test-bed in 15  which the entire system is precisely and accurately modeled. The ensemble climate data forecasts, the hydrological modeling and ESP forecast generation, optimization model and decision-making process are all integrated, as is the simulation model that updates reservoir levels and computes generation at each time step. The study focused on one hydropower system both with and without minimum base load constraints. This study finds that the tested deterministic optimization algorithms lack the capacity to compensate for uncertainty in future inflows and therefore increases the odds of forced spillage by attempting 20  to maximize short-term profit by keeping a higher net head. It is shown that for this particular system, an increase in ESP forecast inflows of approximately 5% allows managing the reservoirs at optimal levels and producing the most energy on average, effectively negating the deterministic model's tendency to underestimate the risk of spilling. Finally, it is shown that implementing minimum load constraints serves as a de facto control on deterministic bias by forcing the system to draw more water from the reservoirs than what the models consider optimal trajectories.

25  # 1 Introduction

Hydropower is one of the most reliable renewable energy sources currently available. Managing a hydropower system can be relatively simple, such as for a single run-of-river generating station; or can be very complex such as when multiple cascading reservoir generating stations are to be operated simultaneously. The optimal management of available and incoming water volumes and the effects on downstream elements of the system must consider many sources of uncertainty. 30  For complex systems, the operational decisions must be made based on inflow forecasts, which contain uncertainty derived




from the hydrological modeling chain. Model initial states, incoming weather data and model structure and parameterization all contribute to the overall uncertainty in the streamflow forecasts (Liu and Gupta, 2007). One way to include the uncertainty in the water resources management process is to work in a probabilistic framework. Extended Streamflow Prediction (ESP), a dynamic method that uses historic climate data as future weather scenarios, was designed to provide

multiple scenarios of possible future inflows for a given initial model state, thus allowing the exploration of possible outcomes instead of a single outcome as would be the case with a deterministic forecast (Day, 1985). Typically, the ESP methodology is implemented for long-term forecasts where numerical weather prediction systems are not reliable (i.e. greater than a few weeks). Recent development in sub-seasonal to seasonal forecasts might eventually replace the ESP method on these timeframes, making the ESP only useful for longer-term forecasts (i.e. more than two or three months).

Ensemble forecast quality can be assessed using many metrics, ranging from the simple Mean Error (ME) to Probability Integral Transform (PIT) histograms (Hamill, 2001) and Reliability Diagrams (RD). The current state of literature suggests that a good ensemble forecasting system must produce ensembles that are unbiased, sharp, and reliable, meaning that the uncertainty in the ensemble faithfully represents the real-world uncertainty. A review of ensemble forecast quality metrics

can be found in Hashino et al. (2002) and an analysis of benchmarking methods is presented in Pappenberger et al. (2015).

Ensemble forecasts can suffer from errors in ensemble mean (bias) and spread (dispersion/variability) when compared to the actual outcomes as measured over a long time period (Wood and Schaake, 2008). The hydrological community has found clever ways to correct bias in ensemble forecasting, such as correcting precipitation amounts of input climate data

(Crochemore et al., 2016; Chen et al., 2014; Voisin et al., 2010; Gneiting et al., 2007) or modifying initial conditions with data assimilation (Liu and Gupta, 2007; DeChant and Moradkhani, 2011). Ensemble forecast spread (including ESP and short-term, weather forecast-driven forecasts) has also been tackled recently by different methods such as pre-processing of inputs (Arsenault et al., 2016) and post-processing of model outputs (Pagano et al., 2013; Boucher et al., 2012, 2015, Hashino et al., 2007). Zhao et al. (2011) analyzed the effect of streamflow forecast uncertainty for reservoir operation for

both deterministic and ensemble forecasts, showing that improved uncertainty representation could lead to better decision-making. Bias-correction of ESP forecasts has been evaluated in Hashino et al. (2007), who report improvements in seasonal volumetric forecast quality with the application of three bias-correction techniques based on transformations derived from historical simulations and observations. Boucher et al. (2012) assessed the economic aspect of hydropower generation using short-term (10-day) ensemble forecasts and found that post-processing the forecasts using the best member method

(Roulston and Smith, 2003)) and a similar method proposed by Fortin et al. (2006) improved reservoir management and energy generation. The post-processing was performed on the entire length of the forecasts. Boucher et al. (2015) then analyzed statistical post-processing methods in short-term ensemble forecasts, namely for bias-correction, on synthetic data generated following normal and gamma distributions. They explored the effects of post-processing on ensemble spread but did not consider the impacts on reservoir management. Anghileri et al. (2016) showed that their one-year climatology-based



ESP forecasts were 35% less informative than perfect forecasts and that improving the ESP forecasts had value on a wide range of reservoir characteristics using seasonal to inter-annual lead-times. Côté and Leconte (2015) studied the impacts of ESP under-dispersion on electricity generation of a hydropower system located in Quebec, Canada, and concluded that under-dispersion in the ESP ensemble negatively impacts the operating policy of the system, but the impacts differ

depending on the optimization algorithms used to derive the policy. In this study, we explicitly analyze the effects of biases in ESP forecasts and verify the robustness of the same hydropower system to these biases. To our knowledge, this is the first attempt to quantify the impacts of long-term ESP forecast biases on a hydropower system's performance, although similar studies have been performed for short-term forecasts (Cassagnole et al., 2017).

This study aims to identify and quantify the effects of ESP forecast bias on the average hydropower output of the Saguenay-Lac-St-Jean (SLSJ) system when managed under different conditions, notably (1) using three optimization and decision-making algorithms and (2) with and without minimum load (or generation) constraints (MLC). In other words, how does a biased ESP affect the reservoir management policy optimization and hydropower generation considering unknown future inflows? Understanding the effects of ESP forecast biases will help quantify the true value of unbiased forecast in a

hydropower generation context and help understand how to maximize generation efficiency and increase expected overall profits. A test-bed that emulates the real-world system and that can be run in hindcast mode was developed to measure the hydropower generation over the past 25 years. This method was selected to limit differences between the various simulations by ensuring that the system is consistent between them.

The next section presents the study area and data, section 3 introduces the methods and models used in this study and section 4 details the obtained results. Sections 5 and 6 respectively contain the discussion and concluding remarks.

## 2 Study area and data

### 2.1 Study area

This study was performed on a hydroelectric system in the province of Québec, Canada. The hydropower system in question,

the Saguenay-Lac-St-Jean (SLSJ) hydropower complex, is wholly owned and operated by Rio Tinto Aluminum's Power Operations (RTA) and is used mainly to supply the large energy requirements of the company's aluminum smelters. On average, the system does not produce enough hydropower to fulfill the energy needs of the smelters, therefore MLCs are imposed to minimize the amount of energy that must be purchased (Arsenault et al., 2013). More details on the operational constraints are presented in section 2.3 and section 3.4.

Rio Tinto holds water rights on a 75000 $km^2$ drainage basin on which six hydroelectric generating stations were built. Four large reservoirs also help in buffering and routing the flows and provide more constant electricity generation. However, they




also add to the complexity of optimizing the water drawdown policies. Figure 1 shows the catchment location as well as the positions of the reservoirs and generating stations. Table 1 gives an overview of the generation capacity and average streamflow at the various sites, and reservoir characteristics are shown in Table 2. In the current operational setting, the SLSJ basin is divided into 5 sub-catchments:

- Lac-Manouane (LM), which contains the Lac-Manouane Reserervoir (RLM) reservoir. RLM is a managed 2657 hm$^3$ reservoir but does not have hydroelectric generation capacity. It is used to rout flows directed to the second reservoir on the Péribonka River, namely Passes-Dangereuses Reservoir (RPD), through the Bonnard Channel which is a controllable spillway.

- Passes-Dangereuses (PD) contains a large reservoir (5227 hm$^3$) which is used to feed the Chute-des-Passes generating
station (CCP). Water drawn from the PD reservoir, either through CCP's turbines or the spillways, then flows to the Chutes-du-Diable sub-catchment.

- Chutes-du-Diable (CD) contains a smaller reservoir (345 hm$^3$) and is largely influenced by water drawdowns from CCP. Its outlet is defined as the Chutes-du-Diable generating station (CCD)'s location. Water then transits to the Chute-à-la-Savane sub-catchment.

- Chute-à-la-Savane (CS) is the last sub-basin on the Péribonka River. A run-of-river generating station, Chutes-à-la-Savane (CCS), defines its outlet.

- Lac-Saint-Jean (LSJ), which is the largest catchment at over 45000 km$^2$. It essentially drains all the other unmanaged rivers to the Lac-Saint-Jean Reservoir (RLSJ) and excludes the Péribonka river system. The RLSJ reservoir is relatively large at 4550 hm$^3$ and it feeds the Isle-Maligne powerplant (CIM).

Water drawn from the RLSJ reservoir is finally routed to two parallel powerplants sharing a reservoir small enough to be considered run-of-river, although hydraulic head is approximately 46 meters at the Chute-à-Caron powerplant (CCC) and approximately 63 meters at the Shipshaw powerplant (CSH) due to the difference in downstream elevations.

## 2.2 Hydrometeorological Data

All data for this study were taken from operational databases. Observed streamflow (and mass-balance derived inflows for
managed sites) were taken from hydrometric gauges owned and operated by RTA. These data were used to calibrate the hydrological model but were otherwise not used in the study. Instead, a proxy for observed streamflow was generated as described in section 3.2. Climate data fed to the hydrologic model, including precipitation and maximum and minimum temperatures, were collected by RTA's private network of 22 weather stations.



Hydrometric data is available from as early as 1916 for the LSJ sub-basin; however only in 1953 were all the other sites gauged and recorded. Climate data is also available starting in 1953, when there were fewer stations, until the present day. A major investment in weather stations was made in 1986, by which point the entire network was up and running as it is today. In all cases, weather data was interpolated over the catchment to drive a distributed hydrological model.

### 2.3 Description of operational constraints and data

All other data used in this study, such as detailed generating station characteristics, operating rules, import/export energy contracts and power contracts reflect the current operational state of the system; however, they are proprietary and cannot be disclosed in this paper. A high-level overview is nonetheless given here to contextualize the problem, and the mathematical description of the problem and optimization models is given in section 3.3. In essence, the hydropower complex is used to generate electricity for aluminum smelters. By the nature of these smelters, the power level must never drop below a certain threshold (minimum load) or the aluminum extraction by electrolysis process could be ruined for whole batches of aluminum product. Therefore, contracts are in place with other utilities as backup to provide power should the generating stations fail to meet demand. Contracts also exist in the other direction, should more power than required be generated. Moreover, the generation planning is largely influenced by seasonal climate variations, with the lowering of water levels in reservoirs during winter and filling during the spring. ESP forecasts are used to estimate the best water drawdown decisions for each day and for each site. This "decision" is a set of streamflow values that must be either drawn from the reservoirs or turbinated at each powerplant for the day. Water levels in the RLSJ reservoir must also be kept within bounds as the reservoir is used for tourism and agriculture on top of hydropower generation. The optimal reservoir management strategy is the one that will allow reducing the overall cost of energy. Also, water spillage has a negative impact on the hydropower production as it increases the tailrace elevation and reduces the net head. Otherwise, spilling is not penalized since sometimes, due to limits on reservoir constraints, the water can add a negative price while water must be spilled to ensure safe management of the reservoir.

### 3 Methods

### 3.1 Current operational setting

The current seasonal water management process of the system uses the traditional ESP forecast approach (Day, 1985) based on a 64-year historical climate record. A hydrological model produces the inflows for a specified duration, normally between 3 and 6 months lead-time for this catchment with long memory due to snow processes. Then, an optimized water release policy is computed for each scenario in the ESP ensemble. A deterministic optimization algorithm that uses inflow scenarios as possible future realizations of the inflow calculates the optimal releases at each site (the set of water releases at all sites is referred to as a "decision"). In the optimization algorithm, a piecewise linear approximation of the hydropower function of



powerhouses is used (Hamann and Hug, 2014), which linearizes the optimization model. One problem with managing reservoirs is that the optimization algorithms attempt to empty the reservoirs at the end of the forecast window because the future value of water in the reservoir is zero unless otherwise specified. Therefore, a water value function was derived for each day of the year (365 values) by using a Sampling Stochastic Dynamic Programming (SSDP) algorithm on historical

inflow data (Faber and Stedinger, 1990; Côté et al., 2011; Côté and Leconte, 2015). This ensures that no matter the forecast data and duration, there is always a value function that can estimate the value of water remaining in the reservoir at the end of the period. These value functions are also approximated by hyperplanes, essentially tangential surfaces to an N-dimensional function. The Fico XPRESS linear solver (FICO, 2017) is used to solve the resulting linear programming model. This allows the management team to assess the possible outcomes and determine if special action is required. Most of

the time, in the absence of particular hydrological or operational conditions, the applied decision is based on the ensemble member with the median inflow volume.

### 3.2 Hydropower system simulator test-bed

In this study, the operational setting was precisely and accurately modelled in a test-bed which allows simulating the historical operation of the hydropower system, as shown in Fig. 2. The cycle in Fig. 2 represents the hydropower simulation

test-bed and its sequential and repeating steps. Each of these steps is described below and in the following sections.

1) The process starts at day $t = 0$ with the assimilation of streamflow data to minimize initial condition bias. In this study, this step was not required because the model simulations are used as proxies for the observed inflows. Therefore, the model in simulation mode is always perfect and does not require data assimilation to be implemented.

2) The ESP forecasts for 120 days are prepared according to the procedure presented in section 3.2.

3) The hydrological model is run for each historical climate scenario, always starting with the same initial conditions.

4) The generated ESP forecast is used to drive the optimization method, which also depends on the initial state of the system (reservoir levels at each site).

5) The optimal decision based on step 4 is selected and implemented, and the system is simulated using this decision. This results in a modification in the reservoir states for the next step as well as in estimations of energy generation for the period

and flow rates at each site for the current time step.

6) Steps 2-5 are repeated for each 3-day time-step on the 25-year simulation period.

Furthermore, in this study, the test-bed was run using ESP forecasts with varying levels of bias. This allowed evaluating the energy generation, spillage and reservoir levels at each time step for a given ensemble inflow forecast dataset, therefore





permitting the quantification of the effects of bias on reservoir management and energy generation. Furthermore, criteria for selecting a decision from the deterministic approach were analyzed by selecting decisions of monotonically increasing percentile values (e.g. 10th, 20th... 90th percentile decision, where a "percentile decision" is defined as the percentile taken in the set of outflow rates for each member for the time period) or of inflow volumes (decision related to the 10th, 20th... 90th

percentile inflow volume scenario). In the first case (percentile decision), the outflow rates are the releases determined by the optimization algorithm. Since it is deterministic and generates optimal release values for each case independently, then there is a set of optimal releases for each scenario in the ensemble. The "outflow rates" are the sums of releases at all sites as determined by the optimization method for the current time-period.

By using the percentile approach, it was possible to evaluate the effect of selecting a member other than the median from

within the ensemble. This method forces a bias by repeatedly selecting higher or lower inflow scenarios over the duration of the 25-year simulation. Conceptually, it is similar to introducing a constant bias in the ESP forecasts and selecting the median.

Finally, once a decision is made, it is applied to the system thus defining the generation of flows and spills at each powerhouse and spillway for the period. A system simulation model evaluates the generated electricity at each site and

updates the reservoir levels. The test-bed then moves forward in time, repeating the process for all periods until the last day for which ESP forecasts are available. It is important to note that the test-bed runs on a 3-day time step to maintain reasonable computing time, and each ESP inflow forecast is 120 days long (40 3-day periods). The average power output from the entire system on all periods is finally computed. The test-bed was run with and without the imposed minimum load constraints to assess the system's sensitivity to these constraints.

Three main steps were required to perform the study: (1) Preparation of ESP forecasts with varying levels of bias, (2) implementation and application of the reservoir management optimization algorithms and (3) simulation in a forward-rolling test-bed.

### 3.3 Preparation of ESP forecasts

An initial set of ESP forecasts (one forecast per 3-day simulation period) was produced by sampling the climatological

record, as proposed by Day (1985). One important consideration is the need to derive adequate hydrological model initial conditions for each forecast, otherwise the initial model error would already contribute to the ESP forecast bias. Therefore, this study uses a proxy for observed streamflow derived from the hydrological model driven by the observed climate data over the entire period. Because the model generated this synthetic streamflow, forcing the initial conditions to be perfect is trivial as all one must do is run the model with the historic climate data once more until the forecast date. The historic

simulated streamflow is considered as the forecast target, bypassing all issues related to the hydrological model's initial



conditions. The method has been used previously, (e.g. in Shukla and Lettenmaier (2011) and Greuell et al. (2016)) where the pseudo-observations are shown to be the best estimate of the true conditions of the catchment. The ESP forecasts were generated following a straightforward procedure:

1) The hydrological model CEQUEAU (Charbonneau et al., 1977) is calibrated on each of the five sub-basins using the
Dynamically Dimensioned Search algorithm (DDS) (Tolson and Shoemaker, 2007) according to the procedure in Arsenault et al. (2014). CEQUEAU is a grid-based distributed model which is set up on a 10km resolution grid on the SLSJ basin. It uses daily gridded temperature and precipitation data as inputs.

2) Once the model is calibrated, a single simulation was performed using the observed climate data for 1953-2016, thus generating the pseudo-observed streamflow time series. The entire matrix of state variables for each simulation period was
also saved for future use. No data assimilation was performed at any of the periods because from this point forward, the model-simulated discharge is used instead of the observed measured flows. Therefore, the model simulation and pseudo-observed flow are always in perfect agreement at each time step.

3) The initial date of the test-bed was selected to be on December 1st, 1990. December 1st corresponds to the first day of the hydrological year, where the model states are completely independent from the previous year's hydrology due to the near-
zero correlation between the model states and the future inflows for the hydrological year. The year 1990 was used to begin the test-bed simulation because it offered a good compromise between access to "historic" climate data (the test-bed is blind to future climate and therefore cannot use it in the ESP forecast, so 1953-1989 gives a reasonable starting point) as well as providing room ahead to run the test-bed for evaluating the method's performance (1990-2016). Consequently, the hydrological model was set-up with its state variables from December 1st, 1990 from the initial simulation (point 2) above).

4) The ESP forecasts were generated for each day. After some tests (not shown here), the forecast length was set to 120 days. It is anticipated that after 120 days of ESP forecasts, the information gain is marginal at best because the produced forecasts for longer-term would follow the observed distribution. Tests showed that fewer than 120 days could see some cases where the climate ensemble does not completely merge with the distribution of actual climate outcomes on this system. The ESP construction begins by identifying climate data series starting on the same day as the test-bed's current day (December 1st in
this example) for each of the years on record-to-date. For example, for a 120-day ESP forecast, member 1 would represent climate data from December 1st, 1953, member 2 from December 1st, 1954, and so on. The final ESP forecast is therefore a 120-day by N-year member ensemble. For the scenario-tree optimization algorithm, it is required that the members of a given ensemble all share the same inflow for the first 3-day period because the first-period decision must be unique (therefore requires the same inflow). The average of the members' first-period inflows was taken, and this value was used
for all members in the ensemble



5) The last step in producing ESP forecasts for this study was to add bias to the ensemble means. To do so, ESP forecast members were multiplied by a factor to shift the distribution upwards (factor > 1) or downwards (factor < 1). The method is the opposite of the Degree-of-Mass-Balance (DMB) method proposed in Bourdin and Stull (2013), where the objective is to remove the bias by multiplying the inflow forecasts by the inflow volume ratios between the ensemble members and the observations. A similar method based on multiplicative linear scaling was used in Li et al. (2018) to bias-correct ensemble streamflow forecasts. In this study, we use the method by forcing an inflow ratio to generate an unconditional bias in the ESP. This allowed adding controlled amounts of bias while ensuring no negative inflow values. Bias factors of 0.93, 0.95, 0.98, 1.02, 1.05 and 1.07 were used, resulting in biases of -7% to +7%. Larger values were excluded because they were not necessary to explore the behaviour of biases on the hydropower system operation. In the test-bed, the inflows correspond to the unbiased (i.e. with no added bias) simulation; consequently, a positive bias would force the observed streamflow in the lower-end of the ESP forecasts.

The ESP forecasts were assessed to identify their biases and reliability on each catchment. Figure 3 shows the Relative Bias (RB), defined as the average difference between the forecast mean and the observations over all forecasts in the sample period. It was measured for each season and is furthermore discriminated according to the observation quantiles to determine conditional bias.

From Fig. 3, it is clear that there are conditional biases in the forecasts. For all catchments and all periods, there is a wet bias for the low-flows and a dry bias for the high-flows. Note that the largest biases occur when forecasting inflows for the lower-percentile actual inflows. The most important dry-biases are found during the high-percentile inflow events during the JJA and SON seasons, which are the periods where the inflows are lowest, thus minimizing the effects of the dry bias to some extent. Also, the CS catchment clearly stands out in terms of bias levels, however it is important to recall that it is by far the smallest one with an area of 1300 km$^2$, which is less than 2% of the size of the entire SLSJ basin. Results for the biased ensembles are not shown as they are simply the same curves shifted by the imposed bias level. Table 3 shows the bias levels for each season as well as the average overall bias for each catchment. Figure 4 presents the Reliability Diagrams which allow estimating the forecast skill and reliability for each season. Note that a perfect forecast would lie directly on the 1:1 line.

The forecasts are generally skillful and reliable. The forecasts remain reliable and skillful for almost all cases except for the LM basin in winter where flows are very low to begin with (DJF, second row in Fig. 4).

**3.3 Implementation of deterministic optimization algorithms for deriving reservoir management policies**

In this study, three optimization methods are used to compute the water releases at the reservoirs (Fig. 5). Each one produces a linear programming model that is solved by the XPRESS linear solver. It is worth mentioning that while more efficient optimization algorithms exist, such as the Stochastic Dynamic Programming (SDP) and variants (Stochastic Dual Dynamic



Programming, Sampling SDP, etc.), implementation can be challenging especially for more complex multi-reservoir systems (Côté and Leconte, 2015). This is why many hydropower utilities and companies still use simpler deterministic methods for day-to-day operations (i.e. as described in Fan et al. (2016)). This study concentrates on deterministic methods only and the results should only be interpreted in this context.

The first model is a deterministic approach where the water release decisions are the ones that are optimized for the inflow sequence that has the median volume. In this case, only one optimization is required to compute the water release decisions. In this study, we analyzed the effects of utilizing a scenario other than the median on the overall generation by taking scenarios based on each of the 10 deciles when ranked according to the average inflow volume for the complete length (40 periods, 120 days) of the scenario. The optimization model solved in this case consists in minimizing the production cost

function in Eq. (1):

$$min \sum \beta a_t - \alpha v_t + \lambda r_t + \eta h_t + y_{T+1}, \tag{1}$$

where $a_t$ and $v_t$ are the energy imports and exports respectively with different prices ($\beta,\alpha$) and $y_{T+1}$ is the value of the water stored at the end of the horizon. The variable $r_t$ is the penalty term for violation of water storage limits at the downstream reservoir and $h_t$ is the penalty for energy shortage (explained later in the model). The water value is defined by a set of

hyperplanes that have been computed with a Stochastic Dynamic Programming solver applied to a simpler and aggregated system of two reservoirs (RLM+RPD and RCD+RLSJ), as in Eq. (2):

$$y_{T+1} \geq \pi_{1,j}\left(s_{LM,T+1} + s_{PD,T+1}\right) + \pi_{1,j}\left(s_{CD,T+1} + s_{LSJ,T+1}\right) + \pi_{0,j} \ \forall j = 1,2,K,J, \tag{2}$$

where $\pi_j$ are the coefficients of hyperplane $j$.

The system is constrained by mass balance equations that describe the dynamics of the reservoirs in series, as shown in Eq.

(3) and Eq. (4):

$$s_{i,t+1} = s_{i,t} + \rho_{i,t} + u_{i-1,t} - u_{i,t} \ \forall i = 2,3,K,N; \ \forall t = 1,2,K,T, \tag{3}$$

$$s_{1,t+1} = s_{1,t} + \rho_{1,t} - u_{i,t} \ \forall t = 1,2,K,T, \tag{4}$$

where $s_{i,t}$ is the water stored at reservoir $i$ at the beginning of period $t$, $\underline{u}_{i,t}$ is the total volume of water released (turbinated and spilled) during period t from reservoir $i$, $\underline{u}_{i-1,t}$ is the total volume of water released during period t from reservoir $i-1$ and $\rho_{i,t}$ is

the volume of natural inflows to reservoir $i$ during the period. Although $u$ includes spills and turbinated water flows, spills only occur when the turbines are at full capacity. The power produced at each power plant is given by a set of hyperplanes and is given by Eq. (5):





$$p_{i,t} \leq \emptyset_{1,i,t,j} s_{i,t} + \emptyset_{2,i,t,j} u_{i,t} + \emptyset_{0,i,t,j} \quad \forall i = 2,3,K,N; \quad \forall t = 1,2,K,T; \quad \forall j = 1,2,K,M_{i,t}, \tag{5}$$

where $p_{i,t}$ is the power produced at power plant $i$ (notice that the LM reservoir has no installed capacity), $\emptyset$ are the coefficients of all hyperplanes and $M_{i,t}$ is the number of hyperplanes for power plant at reservoir $i$ at time $t$. Note that the power production modelling depends on the unit outages at each period and at each power plant which explains the dependence of the period $t$ and power plant $i$ indices on $\emptyset$ and M. The system is also constrained by the load balance equation detailed in Eq. (6):

$$\sum_{i=2}^{N} p_{i,t} + a_t - v_t = \omega_t \quad \forall t = 1,2,K,T, \tag{6}$$

where $\omega_t$ is the load. There is a maximum capacity of energy imports which can be handled by a minimum power generation $\gamma_t$ at each period, shown in Eq. (7):

$$\sum_{i=2}^{N} p_{i,t} + h_t \geq \gamma_t \quad \forall t = 1,2,K,T, \tag{7}$$

In this paper, the MLC refers to the inclusion of this constraint on $\gamma_t$ in the optimization procedure. Finally, there is a minimum storage limit $\delta_t$ at the last downstream reservoir for recreational purposes during summer as shown in Eq. (8):

$$s_{N,t} + r_t \geq \delta_t \quad \forall t \in [t_1, t_2] \tag{8}$$

For the purpose of this study, we suppose that there is no upper limit on water releases at each reservoir which is idealized but quite close to the real operational case based on the hydrological regime in this watershed and the capacity of all spillways, which are almost never fully exploited. To simplify the mathematical notation, we considered here that there are reservoirs at each generating station, but the constraints on upper and lower water levels for the run-of-river stations were forced to be equal. The second method is a modified version of the first method in which each ensemble member is optimized iteratively and independently, returning as many optimal decisions as there are members in the ensemble. We test this approach because it is the one used at Rio Tinto for mostly 20 years. The release decision that is selected in the test bed is the one that releases the median amount of water on the system (total drawdown at all generating stations and spillways). Again, in this study, deciles other than the median were also utilized to evaluate the test-bed's sensitivity to this forced bias. For example, always selecting the 20[th] percentile member should generate results consistent with a dry-biased ensemble forecast.

The third and final approach is a deterministic method based on a scenario tree approach (Carpentier et al., 2013; Fan et al., 2016; Séguin et al., 2016) but incorporates only on branching at the end of the first period. The algorithm computes a single decision at time $t=0$ that maximizes the expected value of future outflows given information from all scenarios. To do so, the deterministic method detailed above computes the values of each scenario individually, which are then multiplied by their





respective probabilities of occurrence. In this study, the probabilities are considered equal therefore all members are weighted equally. The resulting scenario tree, which is a unique first period followed by the individual members, is fed to the linear optimizer which finds the initial decision in such a way that the current benefit plus the expected value for the remaining periods is maximized. Séguin et al. (2017) have shown that this kind of simple tree structure performs at least as

well as a sophisticated scenario tree generation algorithm when compared in a rolling horizon testbed where only the first decision is kept in the simulation process. The optimization problem described in Eq. (1-8) can be modified by including $K$ inflow sequences by Eq. (9):

$$min \frac{1}{K} \sum_{k=1}^{K} \sum_{t=1}^{T} \beta \alpha_{t,k} - \alpha v_{t,k} + \lambda r_{t,k} + \eta h_{t,k} + F_{T+1}(s_{T+1,k}), \qquad (9)$$

Each constraint in the optimization model can be modified in the same way and a nonanticipativity constraint is included to

be sure that the release decision at stage $t$=1 is unique as shown in Eq. (10):

$$u_{i,1,k} - u_{i,1,k-1} = 0 \quad \forall i = 1,2,K,N; \quad \forall k = 2,3,K,K, \qquad (10)$$

Note that this implementation is identical for the water value function where the variable $y_{T+1}$,j now depends on the index of the scenario. Therefore, this approach solves a unique optimization problem at each period and includes all scenarios into a fan structure.

For the SLSJ systems (4 reservoirs and 5 powerhouses), the biggest instance for the linear programming model is composed of 30,000 variables and 160,000 constraints for the largest ESP ensemble, which contains 40 periods of 3-day time steps. This continuous linear programming model is easily solved by XPRESS in less than 3 seconds using the Newton Barrier Method (Wright, 2001). All ensemble members are considered equiprobable.

It is important to recall that even though the optimization methods are deterministic in nature, they are used in a test-bed

containing uncertainty and therefore operate in a stochastic setting, with unknown future inflows.

### 3.4 Minimum load constraints

The entire test-bed simulation was performed with the differently-biased ESP forecasts and for each optimization method. Furthermore, the effects of minimum load constraints (minimum generation that must be maintained to power the smelters) were investigated by running the entire setup twice: once with the imposed constraints and again with the constraints from

Eq. (7) removed.

### 4 Results

### 4.1 Current operational method results





The sensitivity of the operational approach (Optimization method 1) to the inflow percentile selection was first investigated. Results are shown in Fig. 6 for the cases with and without MLC. It is important to note that the actual generation figures are not made available due to their sensitive nature. However, the numbers are presented relative to an arbitrarily fixed baseline value.

From Fig. 6, it can be seen that the MLC significantly change the optimization problem's behavior. For example, in the case with MLC, selecting a member representing a higher percentile in the ensemble decreases the overall performance, whereas in the case without MLC the opposite is true (panel a). In panels b) and c), four percentiles are selected and evaluated for each case. In all cases, the unbiased pseudo-observed streamflow is used as the actual realization. Each year's values are

compared to the long-term average generation figure, resulting in some positive values (better than the long-term average) and some negative values (worse than the long-term average). Note that the units represent average annual efficiency (AAE) (MW/m$^3$/s) relative to the median case. AAE is computed by averaging the period-by-period system efficiency, which is the ratio between the total amount of power generation (MW) and total water discharge (turbinated and spilled water). This ratio allows determining how efficient the system was in transforming hydraulic resources into hydroelectricity. In both cases, the

20$^{th}$ and 80$^{th}$ percentile values are significantly different than the median and confirm that the observed trends truly represent the underlying difference in year-to-year production values and are not a product of a few outlier years (panels b and c).

Furthermore, the MW ratios with MLC in Fig. 6(a) are lower than with the unconstrained system, which is expected due to the reduced degrees of freedom. The MLC force the system to generate energy even in low-head states, reducing the overall efficiency due to lower head and water shortages. On the other hand, the unconstrained method can lower the energy

generation in dryer periods to maintain a more efficient generation profile and minimize water shortages.

### 4.2 Decision-based optimization methods

Figure 7 shows the performance of the second and third optimization methods (unique decision vs median decision), which are based on the information content of the entire ESP forecast rather than that of a single member. It includes the generation values for different levels of bias both without (panel a) and with (panel b) Minimum Load Constraints. Note that the y-axis

values are different for ease of viewing in Fig. 7.

A few interesting points emerge from Fig. 7. First, the elimination of MLC allowed producing approximately 0.5% more energy. This means that it could be possible to perform a cost-benefit analysis to determine if the advantages of increasing total generation outweighs the costs of back-up contracts for when the minimum loads cannot be sustained. However, this is out of the scope of this paper.



Second, the median decision optimization method is clearly inferior to the unique decision method. On average, the unique decision method outperforms the median decision method for all levels of forced bias. While the values seem small (between 0.5% and 1%), it is important to remember that when applied to the absolute energy values, these differences become important enough to justify further investigation.

Another noticeable artifact in Fig. 7 is the larger spread in values between the cases with and without MLC. It would seem that the constraints imposed upon the system make the decision-making process more robust to bias in the ESP forecasts. Also of note is the fact that the unbiased (no added bias) ESP forecast is the optimal set for the system with MLC, whereas a slight positive bias seems to improve the results for the unconstrained system. To provide an explanation to this finding, the average reservoir storage levels in the two head reservoirs (LM and PD), which have the most influence on the system, were

plotted for the different ESP forecast bias levels while using all ensemble members. Only results for the unique decision method are presented in Fig. 8.

It is apparent in Fig. 8 that the MLC impose a higher rate of water drawdown during the winter period (January to June) to meet the power required for the smelting operations (Fig. 8(b)). In doing so, the head is reduced, as is the overall efficiency as compared to the unconstrained system (Fig. 8(a)). In the unconstrained system, the optimization algorithm aims to keep

water levels as high as possible to maximize hydraulic head and energy production, which it is unable to achieve under MLC conditions.

It is important to note that the reservoir levels in Fig. 8 are 25-year averages. Both simulations start with identical reservoir levels and evolve over the 25-year simulation period. Also, it is seen that the +7% bias has lower reservoir levels and the -7% bias has the highest ones. This is counter-intuitive but can be explained by evaluating how the deterministic optimization

algorithm impacts the reservoir levels. In the case of a +7% bias, the expected streamflow is 7% higher than the actual inflows will be. This means that the optimization models try to lower the reservoirs to make room for the higher inflows. Of course, these never materialize and the reservoir level ends in a lower state. The opposite is true for the negative biases, in which the optimization algorithm is tricked into thinking that a dry spell is forthcoming but in fact more water actually proceeds to enter the reservoir, causing an increase in reservoir level and increased spillage.

**5 Discussion**

**5.1 Limitations**

In this paper, it must be acknowledged that the test-bed is a simplified approximation of the real-world system. The results obtained herein can be considered estimates of what an automated decision-making system would return. However, the real system is managed by engineers whose experience can lead them to modify the actual decision to mitigate risk or take

advantage of unusual situations. Nonetheless, as more and more entities manifest interest in an "over-the-loop" forecasting



and decision-making framework (Mendoza et al., 2017; Pagano et al., 2016; Liu and Gupta, 2007), it is imperative that the role of each component be well understood, including that of the optimization algorithms and ESP forecasts. As was shown, in some instances, biases in ESP forecasts can be beneficial and any work to correct this bias could in fact be negatively affecting overall generation performance. This highlights the importance of active collaboration between hydrologists and
operations research specialists in hydropower systems management.

Furthermore, the test-bed is run in three-day increments, with each period being attributed the average value for the three days (inflows, power output and reservoir levels). This aggregation creates situations where normally a spillway would be opened on day two of three, but in the test-bed the decision must be made for the entire period. Therefore, small inefficiencies are introduced. Nonetheless, the tested methods in this paper were all subjected to this constraint, so the results
are still comparable. However, comparing these results with a real-world case would highlight these differences. One way to overcome this problem would be to run the system on a daily time step; however, tests on a shorter period showed that the difference is negligible for our case study. The slight differences between the 3-day and 1-day time steps did not justify the large increase in computing time that would have been necessary.

Finally, in the real-world system, the ESP forecasts can be evaluated for bias and/or dispersion issues with a long enough
historical record. Luckily, RTA's dataset covers more than 60 years, which allowed quantifying the ESP error structure. Other systems with fewer data might not have this luxury and would have to rely on shorter length series to establish ESP forecasts. A caveat to this information is that this study supposes stationary conditions, whereas it is possible that climate change has affected recent years or could affect future years. In all cases, the results seem to show that ESP forecast bias affects management policies and power generation. However, this study was conducted in an environment in which the ESP
forecasts were also well-dispersed. Most water resources systems - including RTA's - use ESP forecasts that are under-dispersed (Pagano et al. 2013, Hamill et al. 2001, Arsenault et al. 2016). The combined effects of bias and under-dispersion have not yet been evaluated and could be the subject of a future study.

## 5.2 Comparison between the optimization approaches

The three optimization approaches were investigated to understand how they are affected by biased inputs. First, from Figs.
5-8, it is clear that the results strongly depend on the application of generation constraints (MLC). The biases clearly affect all methods less when MLC are applied. It seems that the MLC constrain the system enough that impact of the ESP forecast is lessened, thus the biases are also less impactful.

Second, for the first method, i.e. optimization on the median inflow scenario, biasing the ensemble had a direct impact on the results, indicating that the method is less robust to the inputs than stochastic optimization methods. Similar conclusions were
revealed in Fan et al. (2016). The same is true for the second method, i.e using the median decision from a set of deterministically optimized scenarios.



One possible explanation is that while the median decision method uses all scenarios to take the median decision, it must make a compromise to do so and thus it discards the entire information content contained in the other scenarios. The unique decision method, on the other hand, optimizes the entire tree and is therefore more informed than the median decision method. In all cases, the unique decision approach (method 3) seems to be more robust to inflow biases and generally

performs better at maximizing hydropower output. This is to be expected as stochastic methods are known to be more efficient by considering this uncertainty (Faber and Stedinger, 2001).

### 5.3 Effects of deterministic bias

Throughout this study, it was shown that forcing a positive bias on the forecasted inflows helped generate more power even if the actual realized inflows were kept intact. The increases in the relative MW shown in Fig 7(a) for the case without MLC

is due to the "deterministic bias", i.e. the tendency of deterministic methods to be overconfident, introduced by the optimization method. The deterministic bias is also observed in Philbrick and Kitandis (1999) and is a consequence of the optimization model's perfect foresight of the future inflows. The model thus overestimates the capacity of the system to manage the reservoir at a high head without spilling water. In a rolling horizon test bed, high-flow scenarios will cause larger spillage than expected with the deterministic optimization. It follows that a positive bias introduced in the forecast (or

selecting member percentiles > 50%) will dampen this effect and the resulting policy will manage the reservoirs lower while reducing the spillage. A negative bias will exacerbate the effect and generate more spillage. An interesting aspect of the phenomenon observed here is that when the MLC are added, introducing a positive bias in the forecast has a negative effect. First, a negative bias has almost no effect since the MLC forces the system to produce power to maintain the necessary load. However, when increasing the bias over the 50% percentile, the system anticipates large arrivals of water, which, on average,

do not materialize. This means that the system is always in water deficit and since it must maintain loads, it draws water even when the reservoir levels are low which makes the generation inefficient. There are thus more water shortages and power must be purchased to fulfill energy requirements.

The effect of bias levels on the reservoir storages (Fig. 8) is informative because it shows that the optimization methods are directly influenced by the inflow volumes they are given in the ESP forecasts. Larger forecasted inflow volumes correlate to

lower reservoir levels and vice-versa. From the optimization algorithm's point of view, it is better to increase generation despite slightly lower efficiency than to maintain maximum efficiency and then spill the large perceived inflows. Since the actual realized inflows are lower than the ESP forecasts on average, this translates to lower average reservoir levels (and thus lower hydraulic head) but fewer unproductive spills. On the other end of the spectrum, negative biases in the ESP forecasts force the optimization methods into higher reservoir storages to save water and operate at maximum efficiency with the

highest hydraulic head possible. In this case, when the actual inflow materializes, it is on average higher than the anticipated inflows, which makes unproductive spills more frequent. Therefore, low biases can also lead to lower generation figures. These two effects follow from the same deterministic bias as was demonstrated in Fig. 7.



## 5.4 Quantification of deterministic bias uncertainty

These seemingly trivial findings beg the question: if high and low biases are penalizing the generation figures, how can the results in Fig. 7 be explained? Recall that for the unconstrained system, positive biases actually increased overall generation, with a 5% positive bias being the optimum (Fig. 7(a)). It is important to recall that the original forecasting approach produced slightly dry bias (Table 3) of approximately 1-2%. However, the 5% wet bias still outperformed the 2% wet bias trials, indicating that more is at play than the simple bias correction of the original forecasted ensemble.

These results can again be explained by the deterministic nature of the optimization algorithms. In all cases, the optimization algorithms do not consider uncertainty in the ESP forecasts and find the optimal decision according to the predicted inflows. By their very nature, for a given volume of water, they will find the best management policy, which is one that maintains as much head as possible and minimizes spillage. Unfortunately, the actual inflow volumes are sometimes larger than expected, which makes costly spills necessary. For the unconstrained system, by increasing the bias levels slightly, the model will change its behavior to draw more water to reduce the spills it considers inevitable. However, the actual, lesser inflows are then less likely to force unwanted spills. The same holds true for the constrained system, however the MLC are a natural buffer that force the reservoirs into lower storage ranges, thus also minimizing spills. Therefore, the imposed constraints guard against the deterministic optimization method's blindness to uncertainty. While these results are demonstrated on the SLSJ hydropower system, the theory should be applicable to all hydropower systems which are noticeably affected by hydraulic head and that use deterministic optimization algorithms. The optimal levels of bias for unconstrained systems will most probably vary from site to site depending on the hydrological forecast quality and the system's complexity and capacity. Furthermore, an element that must be considered is the size of the reservoirs as compared to the 3-day inflows. Inflow ratios range from 1.1% to 12% of reservoir capacity when using the average inflow values. It is expected that the smaller reservoirs will be more strongly affected by biases in ESP forecast as they could lose efficiency (from net head elevation or by having less time to react to prevent spilling) much quicker than a relatively large reservoir. Fortunately, in this study the most powerful generating stations are backed by large reservoirs or are run of river and are thus not affected by this problem. The CCD generating station is the most vulnerable in this regard and only contributes 235MW out of the total ~3200MW of installed capacity.

## 6 Conclusion

This study aimed to identify the effects of ESP forecast biases on optimization methods used for managing water drawdown policies in a hydroelectric complex. A test-bed simulating the real-world system was set up to identify how three decision-making methods were affected by ESP forecast biases and minimum load constraints. For RTA's SLSJ system, it was possible to identify and quantify the energy gains and losses due to each of these factors. A few key points stand out and

should be kept in mind during the implementation of a forecast-optimization-decision framework in hydropower management.

First, the results tend to indicate that the unique decision algorithm performs better overall than taking the median decision amongst the decisions for all scenarios taken independently. The information content of the entire tree provides better

results than the compromise solution of the median decision and makes better use of a large ESP ensemble.

Second, on systems where there are fewer constraints, a slight positive bias in the ESP forecasts is desirable as it compensates the lack of consideration for uncertainty of the deterministic optimization algorithms, making the entire system more efficient on average. The amount of bias was quantified at 5% for this particular system. Other systems should also behave similarly, with bias levels varying from one site to the next, but this remains to be validated. This study seems to

indicate that a more heavily constrained system would be more robust to bias because of its reduced degrees of freedom, which limits the frequency of full-reservoir states. Logically, a system that would be so constrained as to not be flexible at all (e.g. always produce maximum energy) would not be affected by the forecast biases as the forecasts would play no role in the operating policy.

For the study site, the optimal setup was found to be a unique-decision optimization method with the full ESP forecasts and

with no minimum load constraints. However, operational needs mandate that the MLC be respected, but looking to exchange energy with partners through contracts could be a viable strategy to increase overall power generation and system efficiency. This would inevitably allow generating more energy overall (due to removal of the constraints). The hypothetical question here is "could it be possible to negotiate a contract that allow purchasing more power when in deficit (to cover the minimum loads of the smelters), and would the increased long-term generation be sufficient to cover the cost of such a contract?" This

should be investigated in future work.

While this study looked at deterministic optimization methods, stochastic methods could also be implemented and tested to compare their behavior when subjected to biased inputs. Future work could also analyze the combined effects of bias and under-dispersion used with deterministic and stochastic optimization methods. This would pave the way to better understanding of the desired ESP forecast properties for optimal water drawdown policies for hydropower systems

management.

## 7 Data availability

The climate and flow data can be accessed by writing to the authors and filling a Non-Disclosure Agreement under certain conditions.

## 8 Competing interests





The authors declare that they have no conflict of interest.

## 7 Acknowledgements

The authors would like to thank Kenjy Demeester for his help in extracting and preparing the data. We would also like to thank Rio Tinto for the datasets and models used in this study. Finally, we wish to acknowledge the contributions made by anonymous reviewers who helped shape it in its current form.

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

**Figure 1: Study area location, hydropower generating stations and reservoirs. The inset in the right panel shows the hydropower system schematic.**




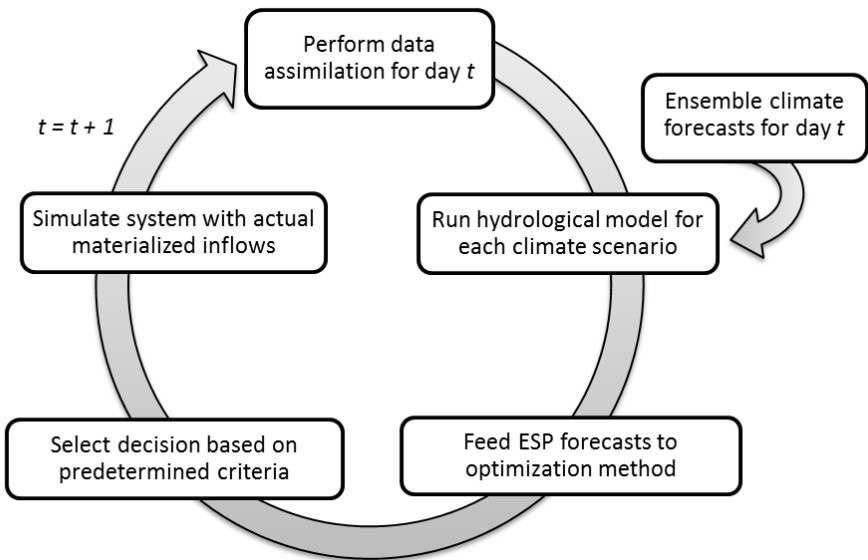

**Figure 2: Over-the-loop system simulation test-bed diagram.**



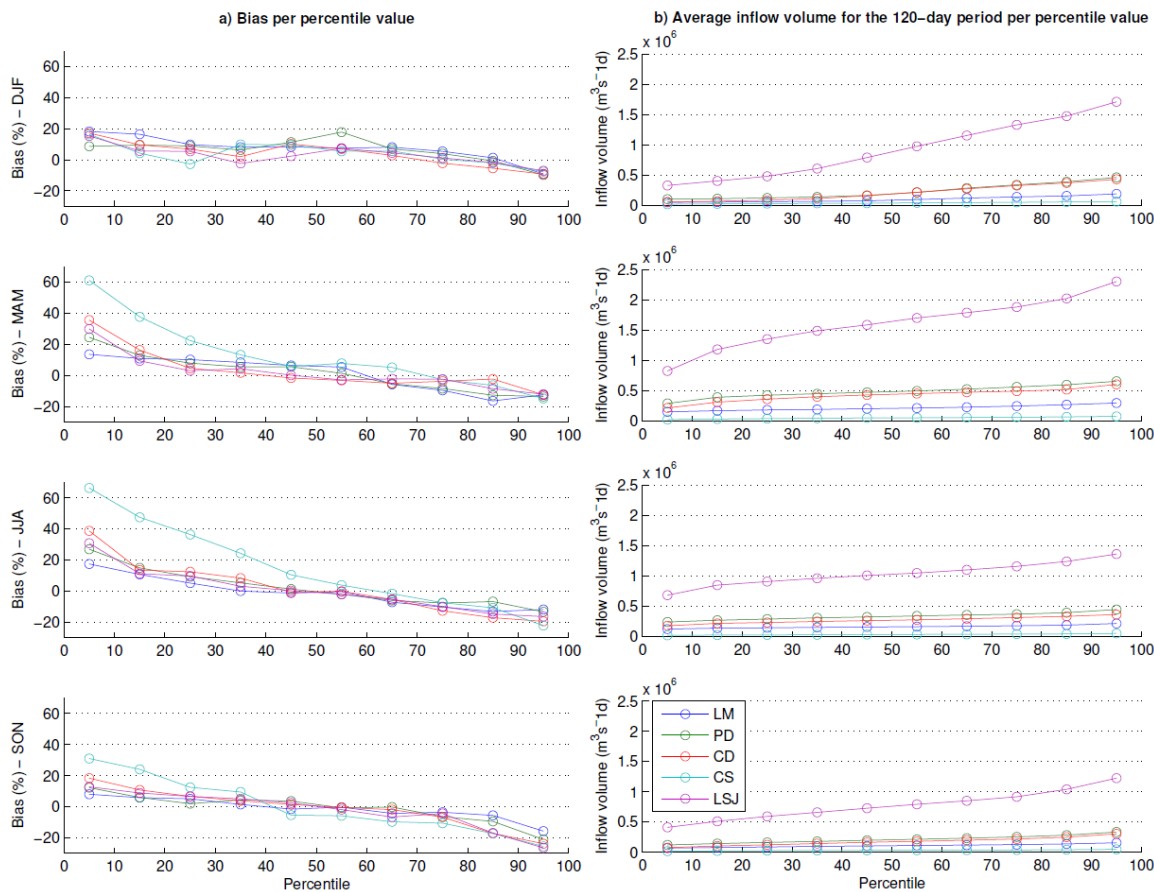

**Figure 3: Bias for each catchment discriminated per season and per percentile of observed flow (column a) and average inflow volume for the 120-day period populating the figure, also discriminated by season (DJF = December, January, February, etc.) and percentile value.**


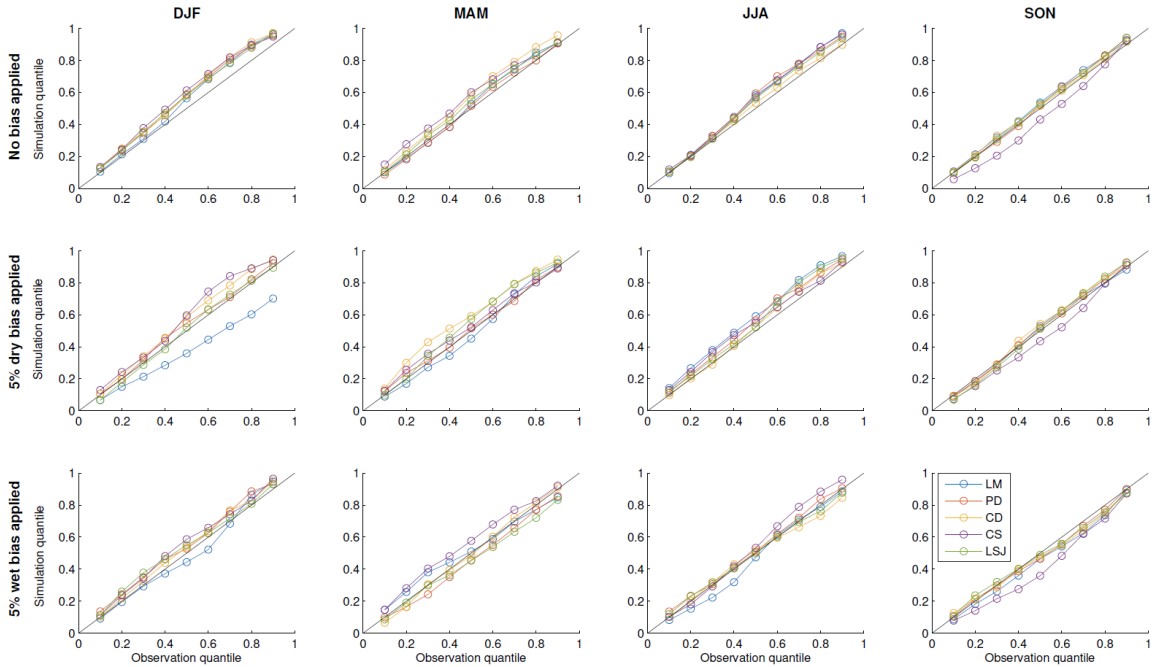

Figure 4: Reliability diagrams for seasonal forecasts. The first row presents the results for the original forecasts, the second row contains a 5% dry bias and the third row includes a 5% wet bias.

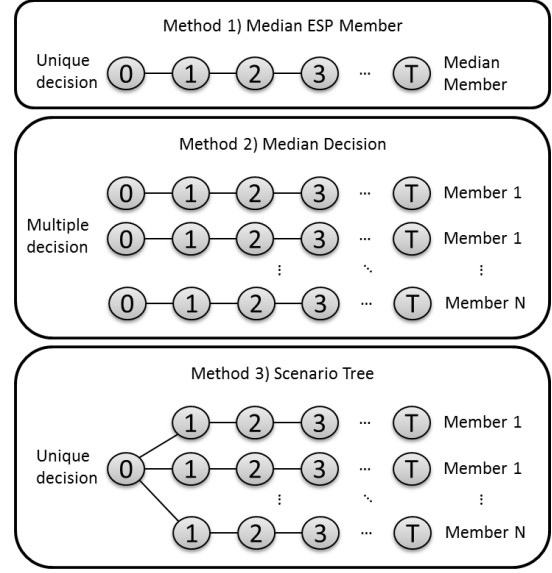

5    Figure 5: Overview of the three optimization algorithms and their decision points.





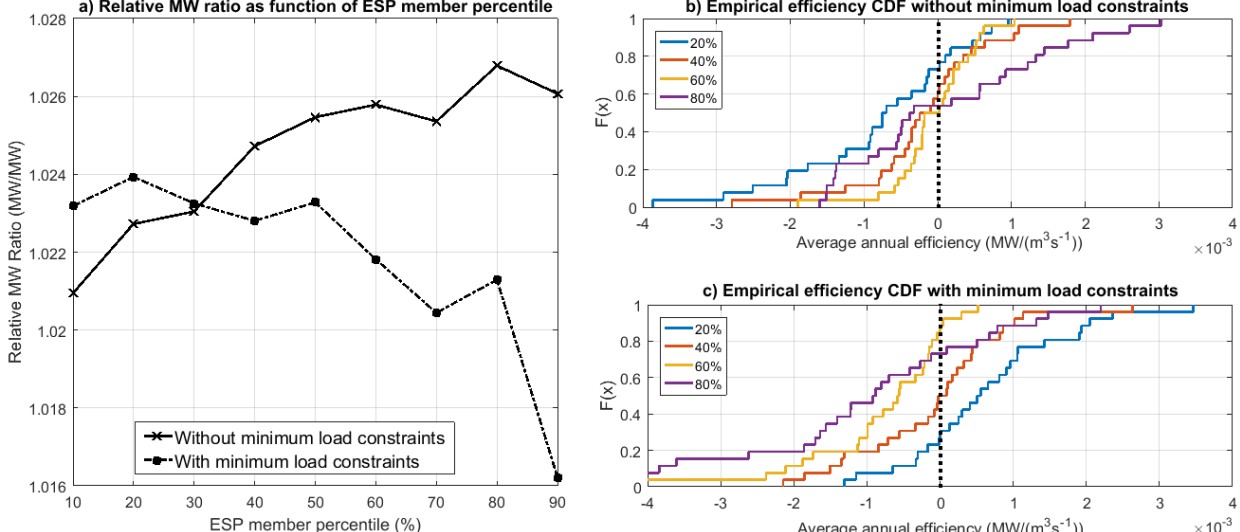

**Figure 6: Average energy generation of the entire system with varying levels of ESP forecast bias when the decision linked to the optimization of the median inflow scenario is used to derive the water drawdown policy. Panel a) shows the relative MW ratio with and without MLC, whereas panels b) and c) show the average annual efficiency for a subset of selected percentiles compared to the median percentile without and with MLC respectively.**





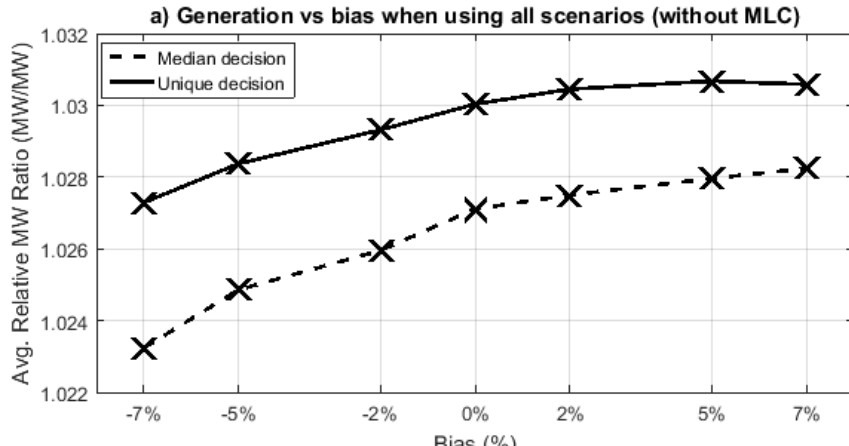

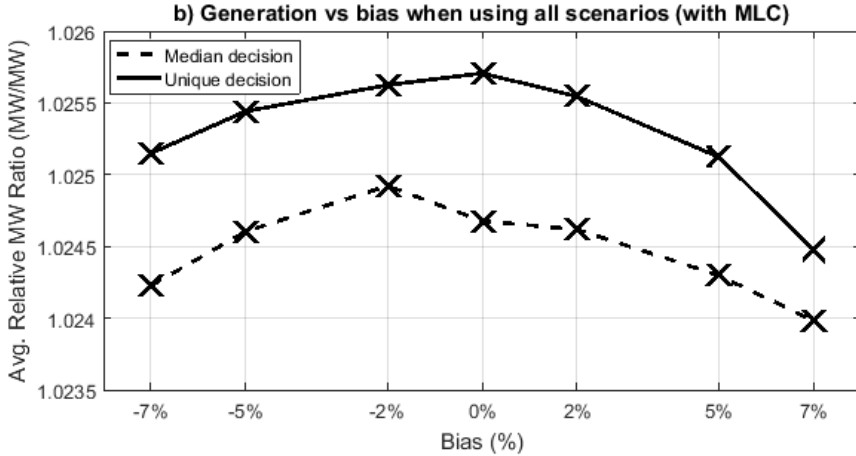

**Figure 7: Average power generation using median-decision and unique-decision optimization algorithms as a function of bias without (panel a) and with (panel b) MLC.**





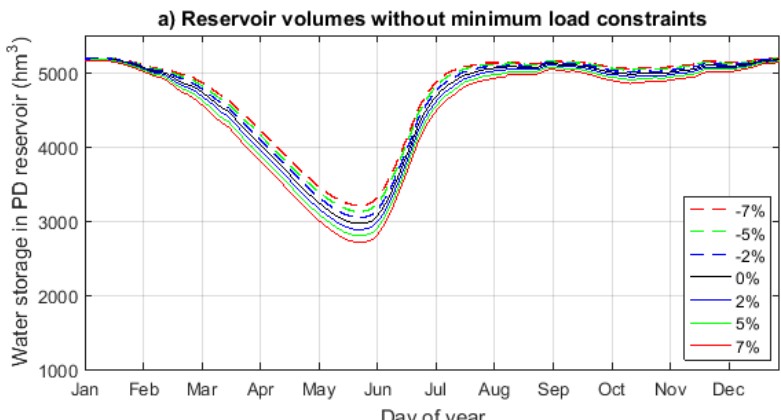

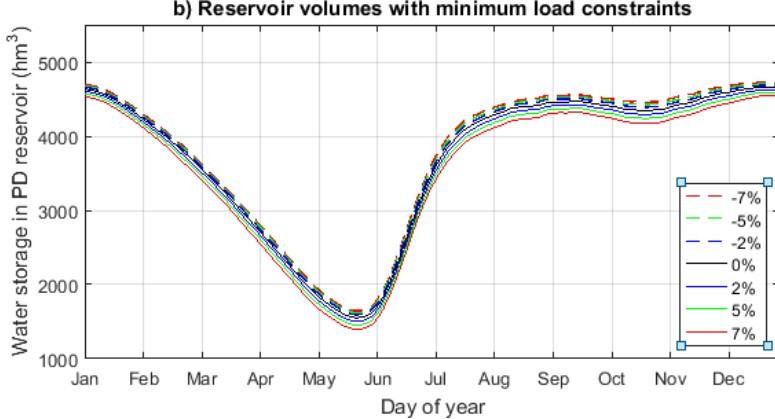

**Figure 8: Average reservoir storage level as a function of time for various bias levels without (panel a) and with (panel b) MLC.**




**Table 1: Generating station and reservoir characteristics on the SLSJ hydroelectric complex.**

| Generating station | Acronym | Installed Capacity (MW) | Maximum hydraulic head (m) | Maximum flowrate before spillage (m3/s) | Reservoir |
|---|---|---|---|---|---|
| Chute-des-Passes | CCP | 844 | 195 | 600 | RPD |
| Chutes-du-Diable | CCD | 235 | 33 | 800 | RCD |
| Chutes-à-la-Savane | CCS | 250 | 33 | 820 | Pondage |
| Isle-Maligne | CIM | 454 | 32 | 1600 | RLSJ |
| Chute-à-Caron/ Shipshaw | CCC/ CSH | 222/ 1172 | 46/ 63 | 2500 | Pondage |

**Table 2: Reservoir characteristics on the SLSJ hydroelectric complex**

| Reservoir identification | Acronym | Average inflow (m³/s) | Storage volume (hm³) | Water level fluctuation (m) |
|---|---|---|---|---|
| Lac-Manouane | RLM | 115 | 2657 | 5.5 |
| Passes-Dangereuses | RPD | 244 | 5227 | 30.5 |
| Chutes-du-Diable | RCD | 166 | 345 | 7.8 |
| Lac-Saint-Jean | RLSJ | 890 | 4550 | 7.9 |

**Table 3: Relative bias between the forecast mean and observed 120-day inflow volumes and classified by the season during which the forecast was made.**

| | Relative bias (%) | | | | |
|---|---|---|---|---|---|
| Catchment | DJF | MAM | JJA | SON | Full year |
| LM | 4.26 | -0.98 | -2.48 | -2.66 | -0.47 |
| PD | 3.33 | -0.61 | 0.20 | -3.69 | -0.19 |
| CD | -0.28 | -0.09 | -1.42 | -5.49 | -1.82 |
| CS | 2.19 | 6.72 | 7.16 | -4.39 | 2.92 |
| LSJ | 0.98 | -0.81 | -1.79 | -5.67 | -1.82 |