# Peer review of "Analysis of the effects of biases in ESP forecasts on electricity production in hydropower reservoir management"

_Hydrology and Earth System Sciences, 2018_

## Referee Comment (RC1) · Anonymous Referee #1 · 16 Aug 2018

\*\*\* General comments \*\*\*

This is a very interesting paper studying, in detail, the effect of forecast bias on electricity production in hydropower reservoir management.

While each and every of the results presented is interesting, I am not convinced by the authors' analysis of the supposedly beneficial effect of a positive streamflow forecast bias on the generation output. While some bias appears to indeed be beneficial for this particular optimization model, it may not be beneficial in general. More on that below.

\*\*\* Specific comments \*\*\*

[Figure]

The comments in this section center on the generation output as a function of stream-flow forecast bias, and on the results leading to Figure 7 in particular.

First of all, some things are not clear to me:

- How is the relative MW ratio computed exactly? Is the result of an open-loop application of optimized reservoir releases to the simulation over the optimization horizon? Or is a closed-loop approach used to produce these figures, where re-optimization is performed at every simulation time step?

- What is the impact of the choice of values for the parameters lambda and eta (Equation 1 on p10)? It seems to me that higher lambda values would also entail more conservative operation and would hence affect the results presented in Figure 7.

The authors point out (p16) the tendency of deterministic methods to be overconfident in their ability to manage a reservoir at high head, thereby causing larger spillage than necessary:

- This will indeed be an issue if the optimization results are applied in an open loop setting. However, if re-optimization is performed every simulation time step in a closed loop setting, the planning will adjust to higher-than-anticipated reservoir levels and spilling should be much reduced.

- Use of a soft upper reservoir water level constraint, rather than a hard constraint, would probably eliminate the spilling issue altogether (in a closed-loop setting).

- With the spillage issue out of the way, the reduced reservoir levels resulting from the positive bias should, in the long run, negatively impact generation output due to a) reduced head and hence reduced efficiency, and b) due to reduced water availability beyond the optimization horizon.

As a result, I am not convinced that the reduced spillage/higher generation output phenomenon is fundamental, and therefore I would suggest to be much more cautious in claiming that a small positive streamflow forecast bias is desirable (p18). Rather,

it strikes me as a phenomenon that emerges out of the interaction between forecast bias and (perhaps, if I understood correctly) the lack of a closed loop, and too stringent reservoir level bound modelling.

*** Technical corrections ***

p7: The need to derive adequate hydrological model initial conditions is pointed out. Then, it is described that these are derived using a hydrological model driven by observed climate data. To me, this begs the question on how this model is then initialized before it is ran "once more until the forecast date"?

p9: It would be helpful to include a formula describing how exactly the relative bias is computed.

p11: Equation 5. I don't see how the fundamentally nonconvex product of discharge Q and head H can be approximated using a set of linear inequalities; consider for example the relation QH restricted to Q=H, this is a convex function, which can – after approximation with a bundle of linear inequalities – be used as a lower bound for the power generation, but not an upper bound (due to the hyperplanes intersecting below the curve). The reverse holds for the relation QH restricted to Q=H_max - H, for example, which is concave. Not sure what the impact of the hyperplane approximation is on your results, but it looks like there will be issues with the head dependence of the power generation. Consider looking at some of the recent work on the homotopy approach towards tackling the QH nonlinearity without sacrificing physical accuracy.

p16: I find referring to the scenario tree approach as being a deterministic approach confusing. Yes, the algorithm is deterministic, but it takes forecast uncertainty into account to some extent and is in that sense probabilistic.

In general, it is also not immediately clear that the "unique decision method" is the same as the "scenario tree approach". Best to make this explicit earlier on.

Figure 6: The units of panels (b) and (c) on the X axis don't make sense to me, esp.

the negative efficiencies.

---

## Referee Comment (RC2) · Anonymous Referee #2 · 30 Dec 2018

This manuscript presents a study on the effects of bias in seasonal forecasts developed using the well-known Ensemble Streamflow Prediction (ESP) approach on release decisions from a series of reservoirs for the generation of hydroelectric power stations. The energy generated is destined primarily for use in the Aluminium smelting processes. This study presents a nice example of the use of seasonal hydrological forecasts in decision making, relating probabilistic forecasts and their typically inherent biases to the decisions that are made with these forecasts. This study is of interest to the readership of HESS, and although I think that the different methods used for the optimising the releases informed by the seasonal forecasts are relevant, I think that the main interest (as also suggested by the title) are in how uncertainties and biases

influence the optimal decisions made. I would, however, suggest some improvements and clarifications to the manuscript to increase the noted appeal to the readership of HESS. One of the main results that the authors seem to conclude is that a forecast without bias is not necessarily as beneficial as when there the forecast has positive bias. To most hydrologists working in seasonal hydrological forecasting, this seems to go against what is often considered as the ultimate goal of bias correction methods: developing an unbiased forecasts. Although the authors elude to it to some extent, one of the main reasons for this being that this tends to avoid spill, which is penalised in the optimisation as the volume of water that is spilled is then not used in generating power. This is particularly so when the minimum base load constraint is not included, as then the optimal solutions then tend to run the reservoirs at low heads (though this will generate less power for the same release discharge, and may incur higher penalties due to the recreation constraint). When the minimum base load constraint this changes, as this is now imposed as a constraint rather than being included as a penalty. Including this constraint reduces the "room" for the optimisation algorithm. I think that this discussion is interesting, but do think it should be generalised in the discussion. The conclusions found are not general to the use of probabilistic forecasts, but are conditional on the shape of the decision making problem (as formulated in the optimisation function). This sheds an interesting light on the value of forecasts, and how value is related to the relative penalties imposed by the different parts of the objective functions (for example recreation versus hydropower generation). I think it would be good if the discussion is extended to reflect how the conclusions found would change if the shape of the objective function changes. What would happen if the hydropower objectives changed (the current requirement would seem to favour a steady load, rather than for example hydro-peaking), how would this change if there were additional constraints or penalties on downstream releases (I would suspect a flood damage penalty would result in the same conclusions as this would also favour spillage being avoided, but an environmental constraint may favour spilling). I think the essence of my comment is that I agree with the authors that it would seem that a biased forecasts is to be preferred but this needs to be considered from the point of view of the decision process that the forecasts are used to inform. From the point of view of the hydrological forecast in its own right it make sense that the forecast is as unbiased as possible. That there is more value in the biased forecast in this case is in essence the result of a transfer of risk through an objective function that is not symmetrical. This risk transfer may work very differently in a different setting. Another good example is in water allocation from reservoirs for downstream irrigation. In this case it would be of value to tend towards a low bias in the ESP ensemble as this avoids the risk of insufficient resource being available to farmers who may have planted on the expectation of a higher availability of water. It may also be interesting to understand if the biases in the ESP forecasts have different effects through the season. I can imagine that the value of the ESP forecast, as well as the effect of biases differs depending on the time of year.

In the paper the approach the authors take is to use the simulated historical discharges as the reference, effectively removing issues with model biases and errors. Though I agree that this is a sound approach, application of the approach in a real situation would mean that these biases and errors should be considered. It would add to the paper if this is discussed. What are the additional biases that would then need to be taken into consideration?

Specific comments: Page 2; Lines 5-10: This section discusses the usefulness of ESP forecasts, commenting that these are particularly of value in the longer term. I think this discussion should be qualified to some extent. The skill of ESP when using climatology as a reference is derived from the persistence of the initial states. For hydrological systems that have a strong nival regime, such as in the case study presented here, there may be such a persistence. This also implies that the skill varies seasonally, being highest at the start of the snowmelt season, and lowest at the start of the snow accumulation season. This seems to be eluded to later (page 8, line 15) where the near zero correlation between the model states and future inflows is mentioned. This would mean that at this point the ESP forecast is solely based on climatology, which

means that there is no skill.

Page 7; Line 30: The authors suggest in the discussion on considering the simulated discharge as forecast target that this by-passes issues related to the initial state of the model. I think it would be more appropriate to refer to this as bypassing issues with model error and how representative the initial state in the model is of the true hydrological conditions at the start of the forecast.

Page 8; line 17-20: The construction of the ESP forecast is discussed where the available data is divided into two periods, with the test-bed run for the second period using the ESP ensemble in the second period. Later in the paper the authors elude to possible issues as a results of the inhomogeneity between these two periods. I think it is relevant to explore this further. Is there a bias introduced due to the selection of the periods, and if so, what is the sign of this bias and how does this compare to the multiplier applied to the ESP to produce biased forecasts. Page 10; Line 21: I found the notation in the equations somewhat confusing. It is not so clear what K. T, J, pertain to. It would be useful if these could be explained. Page 11; Line 4: Unit outages are mentioned, but little detail is provided of what the implications of these outages are. To what extent do these restrict the available options for meeting the objectives? How are these outages determined? Is this according to a set schedule or according to historical data? Page 11; Line 26: "incorporates on branching" – this is a strange formulation; perhaps rephrase.

Page 24; Figure 3: The scaling of the b) figure does not readily help its interpretation. Clearly the LSJ is the larger catchment and hence much larger inflows. Perhaps use relative volumes, with an indication of what the average volumes is in each Is the flow in the left column the mean across the 3-month season? If so, then is this not the same as the volume (the only difference being the multiplication with time).

---

## Author Comment (AC2) · 14 Jan 2019

Please see the supplementary .pdf file for the author response to the review, thank you.

Please also note the supplement to this comment:
https://www.hydrol-earth-syst-sci-discuss.net/hess-2018-236/hess-2018-236-AC2-supplement.pdf
* * *
[Figure]

[Figure]

Fig. 1.

---

## Author Response (AR1)

**Anonymous Referee #1**

*** General comments ***

This is a very interesting paper studying, in detail, the effect of forecast bias on electricity production in hydropower reservoir management.

While each and every of the results presented is interesting, I am not convinced by the authors' analysis of the supposedly beneficial effect of a positive streamflow forecast bias on the generation output. While some bias
appears to indeed be beneficial for this particular optimization model, it may not be beneficial in general. More on that below.

Thank you for these comments. Below we address the specific points raised by anonymous referee #1 and indicate how we updated the paper to reflect the changes. To be clear, we do not advocate aiming for biased
forecasts; instead, we want to show that the value (or impacts) of forecast bias can change dramatically based on the hydropower system setup and constraints.

*** Specific comments ***

The comments in this section center on the generation output as a function of streamflow forecast bias, and on
the results leading to Figure 7 in particular. First of all, some things are not clear to me:

- How is the relative MW ratio computed exactly? Is the result of an open-loop application of optimized reservoir releases to the simulation over the optimization horizon? Or is a closed-loop approach used to produce these figures, where re-optimization is performed at every simulation time step?
In our study, the optimization is a closed-loop system, as shown in figure 2. There is a re-optimization at each step, with the period's generated ESP forecasts. The decision for the first period is then used to simulate the state of the system for the next period, including reservoir levels, spills and generated energy along the system during the period. We then take the average MW generation of all periods and compare that to our baseline value to transform the absolute to relative MW (MW ratio).

- What is the impact of the choice of values for the parameters lambda and eta (Equation 1 on p10)? It seems to me that higher lambda values would also entail more conservative operation and would hence affect the results presented in Figure 7.
Yes, the weights play a role in the model behaviour, but not necessarily as stated above. A higher value of
Lambda would penalize energy shortages, and because the reservoirs are finite and some inflow scenarios are quite dry, it is impossible to always respect this constraint. Maximum levels are hard constraints due to the physical assets that simply cannot be overtopped. Minimum levels are dictated by the water intakes. A higher Eta value, which penalizes constraint violations to summer environmental flows, would penalize water shortages and would thus encourage the model to keep more water reserves for the summer months.

In addition, given the fact that some constraints are impossible to respect 100% of the time, we can follow two options. The first is to give very high penalties to the violation of constraints and then measure how that impacts energy generation. The second method is to try and balance the penalties so the output of the simulation has similar statistics to the historical operation of the system. In our case,   the parameter values were selected as they provided similar energy and reservoir levels (and constraint violations) as were observed in the historical dataset.

The authors point out (p16) the tendency of deterministic methods to be overconfident in their ability to manage a reservoir at high head, thereby causing larger spillage than necessary:

- This will indeed be an issue if the optimization results are applied in an open loop setting. However, if re-optimization is performed every simulation time step in a closed loop setting, the planning will adjust to higher-than-anticipated reservoir levels and spilling should be much reduced.
This is true indeed, and tests with an open-loop optimization show that the spills are much larger than what we see using the closed-loop optimization. However, the closed-loop optimization still uses the entire forecast duration to determine the best decision at the current time step. Even in a closed-loop system, the optimization model determines the best course of action given what it thinks is a perfect forecast and will then attempt to keep reservoir levels as high as possible, to later release the water at the last possible moment to maximize the duration of the higher efficiency turbinating. Then, when the future inflows are higher than anticipated at the next time step, the optimization model must spill the excess water but will only spill the amount it thinks is required to keep the maximum efficiency, which leads it to again be in a higher than desirable state. This all comes back to the fact that the deterministic methods do not consider uncertainty in their decisions, which leads to overconfident decision-making.

- Use of a soft upper reservoir water level constraint, rather than a hard constraint, would probably eliminate the spilling issue altogether (in a closed-loop setting).
As stated above, the upper water level constraint is a hard constraint as it is related to dam safety. We could model the system with some tolerance but this would go against the operational policy. When the limit is reached, there is a mandatory spilling operation to ensure the level does not rise further. These were all performed in a closed-loop setting.

- With the spillage issue out of the way, the reduced reservoir levels resulting from the positive bias should, in the long run, negatively impact generation output due to a) reduced head and hence reduced efficiency, and b) due to reduced water availability beyond the optimization horizon.
We understand the rationale, but we disagree with this statement. The spillage we see is the result of being overconfident in the future inflow forecasts, and as long as there is a gain to be made to creep closer to the reservoir maximum limits, the model will do so. Of course, if we penalize the reservoir limits too much, then the risk will be reduced but the model will be too conservative.

As a result, I am not convinced that the reduced spillage/higher generation output phenomenon is fundamental, and therefore I would suggest to be much more cautious in claiming that a small positive streamflow forecast bias is desirable (p18). Rather, it strikes me as a phenomenon that emerges out of the interaction between forecast bias and (perhaps, if I understood correctly) the lack of a closed loop, and too stringent reservoir level bound modelling.

We agree that the statement about having a forecast bias being desirable is misleading. What we meant is that in this case, we can factually demonstrate that a positive bias improves the results, due entirely to the fact that the optimization methods are imperfect. Therefore, we want to show that in these cases, the optimization methods are biased and that using an oppositely biased forecast can help correct the optimization model's shortcomings. Of course, we think everyone agrees that what we need to target as a community is having unbiased optimization methods with unbiased forecasts. With this paper, we want to draw attention to the fact that there is indeed a risk in using these methods which are imperfect.

Accordingly, we modified the paper to ensure that it reflects these thoughts. We had it read by an independent researcher to ensure that the message we are trying to convey was understood correctly.

*** Technical corrections ***

p7: The need to derive adequate hydrological model initial conditions is pointed out. Then, it is described that these are derived using a hydrological model driven by observed climate data. To me, this begs the question on how this model is then initialized before it is ran "once more until the forecast date"?
We start the model "empty", i.e. with no water in any of the water stores. After one year, the model spin-up is complete. When we perform a forecast, we can simply take the current day's initial states or re-run the model starting from day 1 (empty reservoirs) and let the model arrive to the current day, thus regenerating the model states on-the-fly. We changed the text accordingly.

p9: It would be helpful to include a formula describing how exactly the relative bias is computed.

Indeed, this is a good idea. We have added the equation as Eq. (1) in page 9 and updated the other equations accordingly.

p11: Equation 5. I don't see how the fundamentally nonconvex product of discharge Q and head H can be approximated using a set of linear inequalities; consider for example the relation QH restricted to Q=H, this is a convex function, which can – after approximation with a bundle of linear inequalities – be used as a lower bound for the power generation, but not an upper bound (due to the hyperplanes intersecting below the curve). The reverse holds for the relation QH restricted to Q=H_max - H, for example, which is concave. Not sure what the impact of the hyperplane approximation is on your results, but it looks like there will be issues with the head dependence of the power generation. Consider looking at some of the recent work on the homotopy approach towards tackling the QH nonlinearity without sacrificing physical accuracy.

First, thanks for the information about homotopy applied to this kind of problem. We will consider digging deeper in this aspect in future works, we hope you understand that we consider modifying our entire approach out of the scope of this study.

Second, the modelling of the Q-H is a convex piecewise linear approximation. We take the lower bound of the envelope of the hyperplanes to approximate the real Q-H-P surface, which is convex. It is important to note, however, that in our case study (and quite probably that this is not the case for all systems) the approximation errors of the efficiency are typically found in the very-low Q section of the surface, when Q rapidly transitions from low to high values. In those cases, there can be a small error on H due to the downstream levels affecting net head.. However, for the vast majority of the historical dataset, the operations fluctuate between max flow and approximately half of that. The most critical errors are those found in the high flow cases, but these are very well modelled in the system which uses non-linear functions to best approximate the change in efficiency during transitions in those ranges. Therefore, we are rarely in the lower bounds of the operating policy, which keeps us away from the "problematic" non-convex parts of the problem.

p16: I find referring to the scenario tree approach as being a deterministic approach confusing. Yes, the algorithm is deterministic, but it takes forecast uncertainty into account to some extent and is in that sense probabilistic.

Yes, we agree. We tried explaining that we are using deterministic methods in a stochastic setting, but this phrasing is not clear we modified the sentence to read:
While the algorithm is deterministic in that it returns the same response to the identical inputs, it does make use of multiple scenarios and in that sense, it can be seen as probabilistic.

In general, it is also not immediately clear that the "unique decision method" is the same as the "scenario tree approach". Best to make this explicit earlier on.

We agree, we changed the occurrences of "scenario tree" to "unique decision method". We still use scenario fan to explain that there is a fan of scenarios, but not in the sense of the "scenario tree optimisation" method.

Figure 6: The units of panels (b) and (c) on the X axis don't make sense to me, esp. the negative efficiencies.

The efficiency is measured as the Energy output per volume of turbinated flow, i.e. MW/(m3/s). The higher the number, the more energy is being produced for the same amount of water. The values in panels b) and c) of figure 6 are relative values, i.e. the value of the 80th percentile (for example) as compared to the value of the 50th percentile member. Therefore negative values indicate that those years performed less well (in terms of efficiency) than the median case, and positive values indicate better efficiency as compared to the median case. We changed the axis labels to indicate that it is in fact "Difference in average annual efficiency". Thank you for catching this.

This manuscript presents a study on the effects of bias in seasonal forecasts developed using the well-known Ensemble Streamflow Prediction (ESP) approach on release decisions from a series of reservoirs for the
generation of hydroelectric power stations. The energy generated is destined primarily for use in the Aluminium smelting processes. This study presents a nice example of the use of seasonal hydrological forecasts in decision-making, relating probabilistic forecasts and their typically inherent biases to the decisions that are made with these forecasts.

This study is of interest to the readership of HESS, and although I think that the different methods used for the optimising the releases informed by the seasonal forecasts are relevant, I think that the main interest (as also suggested by the title) are in how uncertainties and biases influence the optimal decisions made. I would, however, suggest some improvements and clarifications to the manuscript to increase the noted appeal to the readership of HESS. One of the main results that the authors seem to conclude is that a forecast without bias is
not necessarily as beneficial as when there the forecast has positive bias. To most hydrologists working in seasonal hydrological forecasting, this seems to go against what is often considered as the ultimate goal of bias correction methods: developing an unbiased forecasts. Although the authors elude to it to some extent, one of the main reasons for this being that this tends to avoid spill, which is penalised in the optimisation as the volume of water that is spilled is then not used in generating power.

This is particularly so when the minimum base load constraint is not included, as then the optimal solutions then tend to run the reservoirs at low heads (though this will generate less power for the same release discharge, and may incur higher penalties due to the recreation constraint). When the minimum base load constraint this changes, as this is now imposed as a constraint rather than being included as a penalty. Including
this constraint reduces the "room" for the optimisation algorithm. I think that this discussion is interesting, but do think it should be generalised in the discussion. The conclusions found are not general to the use of probabilistic forecasts, but are conditional on the shape of the decision making problem (as formulated in the optimization function). This sheds an interesting light on the value of forecasts, and how value is related to the relative penalties imposed by the different parts of the objective functions (for example recreation versus
hydropower generation).

I think it would be good if the discussion is extended to reflect how the conclusions found would change if the shape of the objective function changes. What would happen if the hydropower objectives changed (the current requirement would seem to favour a steady load, rather than for example hydro-peaking), how would
this change if there were additional constraints or penalties on downstream releases (I would suspect a flood damage penalty would result in the same conclusions as this would also favour spillage being avoided, but an environmental constraint may favour spilling). I think the essence of my comment is that I agree with the authors that it would seem that a biased forecasts is to be preferred but this needs to be considered from the point of view of the decision process that the forecasts are used to inform. From the point of view of the hydrological forecast in its own right it make sense that the forecast is as unbiased as possible. That there is more value in the biased forecast in this case is in essence the result of a transfer of risk through an objective function that is not symmetrical. This risk transfer may work very differently in a different setting. Another good example is in water allocation from reservoirs for downstream irrigation. In this case it would be of value to tend towards a low bias in the ESP ensemble as this avoids the risk of insufficient resource being available to farmers who may have planted on the expectation of a higher availability of water. It may also be interesting to understand if the biases in the ESP forecasts have different effects through the season. I can imagine that the value of the ESP forecast, as well as the effect of biases differs depending on the time of year.

We would like to thank the reviewer for this comment. The analysis is 100% correct. Our work shows that the impact of bias in the forecasts is very dependent on the optimization problem (i.e. the response surface of the objective-function). We have modified the discussion to make it clearer that this is the case, and we have given a few examples that we think could lead to similar results. We added the following sentences in section 5.2:

In essence, the more flexibility the system has, the more it can try to leverage high head (which leads to higher performance but more spilling due to the lack of information about uncertainty). Any constraint or penalty that limits this flexibility by forcing lower reservoir levels (such as high costs of flooding, environmental flows or required base load generation) removes the option of over-filling the reservoir and thus leads to less spills. Therefore, the value of forecast quality is highly dependent on the objective function used in the management process. In a highly constrained system, the value of generating unbiased forecasts can be essentially zero. Conversely, a system with few constraints can make use of better forecasts but then the optimization method must also not introduce new biases as was shown in this paper.

In the paper the approach the authors take is to use the simulated historical discharges as the reference, effectively removing issues with model biases and errors. Though I agree that this is a sound approach, application of the approach in a real situation would mean that these biases and errors should be considered. It would add to the paper if this is discussed. What are the additional biases that would then need to be taken into consideration?

We have added a few sentences in section 5.1. Limitations:

In the same vein, the simplification of the system is reflected in the hydrological model, which would normally introduce new biases. The methodology used herein eliminates this bias, but it would need to be taken into account in a real-world application.

Specific comments: Page 2; Lines 5-10: This section discusses the usefulness of ESP forecasts, commenting that these are particularly of value in the longer term. I think this discussion should be qualified to some extent. The skill of ESP when using climatology as a reference is derived from the persistence of the initial states. For hydrological systems that have a strong nival regime, such as in the case study presented here, there may be such a persistence. This also implies that the skill varies seasonally, being highest at the start of the snowmelt season, and lowest at the start of the snow accumulation season. This seems to be eluded to later (page 8, line 15) where the near zero correlation between the model states and future inflows is mentioned. This would mean that at this point the ESP forecast is solely based on climatology, which means that there is no skill.

This is a good distinction, we have clarified it in the text:

The skill of ESP is largely based on the persistence of the initial states, which themselves depend on the dominating processes. In the case of snowmelt-dominated catchments, ESP forecasts can be relatively skillful at the beginning of the snowmelt period, whereas at the beginning of the snow accumulation phase, the initial states play essentially no role on long-term inflow forecasts meaning that the ESP skill is entirely based on the climatology (Harrigan et al., 2018).

Page 7; Line 30: The authors suggest in the discussion on considering the simulated discharge as forecast target that this by-passes issues related to the initial state of the model. I think it would be more appropriate to refer to this as bypassing issues with model error and how representative the initial state in the model is of the true hydrological conditions at the start of the forecast.

Thank you, this is a good point and we have added this detail in the sentence:

The historic simulated streamflow is considered as the forecast target, bypassing all issues related to the hydrological model's errors and its representation of the initial conditions as compared to the true hydrological conditions at the start of the forecast.

Page 8; line 17-20: The construction of the ESP forecast is discussed where the available data is divided into two periods, with the test-bed run for the second period using the ESP ensemble in the second period. Later in the paper the authors elude to possible issues as a results of the inhomogeneity between these two periods. I think it is relevant to explore this further. Is there a bias introduced due to the selection of the periods, and if so, what is the sign of this bias and how does this compare to the multiplier applied to the ESP to produce biased forecasts.

This is an excellent point that we have investigated further. We have added the following lines to section 5.1, 3rd paragraph:

In this study, average inflow volumes between the calibration period and simulation period over the entire system differ by less than 2% (1405 $m^3$/s in calibration, 1432 $m^3$/s in simulation). This could explain a small portion of the results, however the differences are mostly found during the snowmelt period during which there are spills (and maximum generation), limiting the impacts on the rest of the year where the optimization problem is more difficult. There is no doubt however that during other periods, the fact that there is more water in simulation than in calibration would point to positively biased forecast being more efficient.

Page 10; Line 21: I found the notation in the equations somewhat confusing. It is not so clear what K. T, J, pertain to. It would be useful if these could be explained.

We have added the lacking information. First, K is a mathtype translation error, it should read "…" in the series from "1, 2, …, J" for example, instead of "1, 2, K, J". This has been changed in the manuscript, so K is no longer an issue. T is the total number of time steps, and J is the total number of hyperplanes used to approximate the water-value function.

Page 11; Line 4: Unit outages are mentioned, but little detail is provided of what the implications of these outages are. To what extent do these restrict the available options for meeting the objectives? How are these outages determined? Is this according to a set schedule or according to historical data?

We have added the following sentence, referring to the set schedule of unit outages:
"These shortages were forced according to the average outages seen at the different powerhouses, either due to planned maintenance or unit failures. An average calendar was used due to the fact that maintenance outages are relatively simple to reschedule and compose the bulk of unit outages in this system."

Page 11; Line 26: "incorporates on branching" – this is a strange formulation; perhaps rephrase.

Thank you, this was a typo. It should have read "…incorporates only one branching…". It has been changed in the manuscript.

Page 24; Figure 3: The scaling of the b) figure does not readily help its interpretation. Clearly the LSJ is the larger catchment and hence much larger inflows. Perhaps use relative volumes, with an indication of what the average volumes is in each. Is the flow in the left column the mean across the 3-month season? If so, then is this not the same as the volume (the only difference being the multiplication with time).

We have modified the figure to indicate relative inflows in the right column. We also indicated the average values in the figures to be able to reconstruct the absolute values. As for the left column, the units are percentages of relative bias, therefore the use of average inflow or total volume does not change the interpretation of the figure.

[revised manuscript text omitted]